# Loss of Cdc42 leads to defects in synaptic plasticity and remote memory recall

Il Hwan Kim[1][†], Hong Wang[2][†], Scott H Soderling[1,2]*, Ryohei Yasuda[2,3]*

[1]Department of Cell Biology, Duke University Medical School, Durham, United States; [2]Department of Neurobiology, Duke University Medical School, Durham, United States; [3]Max Planck Florida Institute, Jupiter, United States

**Abstract** Cdc42 is a signaling protein important for reorganization of actin cytoskeleton and morphogenesis of cells. However, the functional role of Cdc42 in synaptic plasticity and in behaviors such as learning and memory are not well understood. Here we report that postnatal forebrain deletion of Cdc42 leads to deficits in synaptic plasticity and in remote memory recall using conditional knockout of Cdc42. We found that deletion of Cdc42 impaired LTP in the Schaffer collateral synapses and postsynaptic structural plasticity of dendritic spines in CA1 pyramidal neurons in the hippocampus. Additionally, loss of Cdc42 did not affect memory acquisition, but instead significantly impaired remote memory recall. Together these results indicate that the postnatal functions of Cdc42 may be crucial for the synaptic plasticity in hippocampal neurons, which contribute to the capacity for remote memory recall.

*For correspondence: scott.soderling@dm.duke.edu (SHS); ryohei.yasuda@mpfi.org (RY)

[†]These authors contributed equally to this work

**Competing interests:** The authors declare that no competing interests exist.

**Reviewing editor**: Eunjoon Kim, Korea Advanced Institute of Science and Technology, South Korea

## Introduction

The synapse is a highly dynamic structure exhibiting constant turn over and remodeling. This synaptic morphoring is driven by fibrous actin (F-actin), which creates the underlying cytoskeletal scaffold for neuronal structures (*Dillon and Goda, 2005*; *Korobova and Svitkina, 2010*; *Koleske, 2013*). Particularly, the dynamic actin turnover in dendritic spines, the postsynaptic portion of the excitatory synapse, produces morphological and functional changes in synapses (*Matus, 1999*; *Star et al., 2002*; *Kim et al., 2013*), that are thought to be a fundamental basis of synaptic plasticity underlying learning and memory (*Kim and Lisman, 1999*; *Krucker et al., 2000*; *Fukazawa et al., 2003*; *Okamoto et al., 2004*; *Kim et al., 2013*). Within the spine, small GTPases of the Rho family, such as Cdc42, Rac, and RhoA exert distinct roles for actin remodeling by regulating actin organization by regulating many downstream factors including WAVE, WASP, Arp2/3 and cofilin (*Hall, 1998*; *Jaffe and Hall, 2005*; *Soderling et al., 2007*). In vitro studies have shown that both Cdc42 and Rac promote the formation and maintenance of dendritic spines, whereas RhoA may negatively regulate spinogenesis (*Nakayama et al., 2000*; *Scott et al., 2003*; *Ahnert-Hilger et al., 2004*; *Newey et al., 2005*). These GTPases are regulated by more than 60 activators (guanine nucleotide exchange factors or GEFs) and inactivators (GTPase activating proteins or GAPs) (*Van Aelst and D'Souza-Schorey, 1997*; *Saneyoshi et al., 2010*). Previous studies using two-photon fluorescence lifetime imaging microscopy (2pFLIM) have demonstrated continuous activation of Cdc42 for more than 30 min within single dendritic spines undergoing structural plasticity. This activation is restricted to the stimulated spine heads and shows a steep signal gradient of active Cdc42 at the spine necks (*Murakoshi et al., 2011*; *Yasuda and Murakoshi, 2011*), suggesting that Cdc42 may be intimately involved in long-term maintenance of structural spine plasticity during the sustained phase of spine enlargement.

It is generally believed that the Cdc42 pathway plays a key role in neurite outgrowth (*Mueller, 1999*; *Luo, 2000*; *Aoki et al., 2004*), neuronal polarity (*Schwamborn and Puschel, 2004*; *Garvalov et al., 2007*), neuronal migration (*Wong et al., 2001*), and dendritic morphogenesis (*Scott et al., 2003*)

**eLife digest** Neurons communicate with one another at junctions called synapses, which are typically formed between the dendrite of one neuron and the axon terminus of another. The dendrites are protrusions coming out of the cell body that receive inputs from other cells; the axon is a cable-like structure that enables neurons to contact other cells. In excitatory neurons in part of the brain called the hippocampus, the dendrites are themselves covered in structures called spines, so most synapses are formed between an axon terminus (belonging to the presynaptic cell) and a dendritic spine (on the postsynaptic cell). The hippocampus is necessary for the formation of long-term memories.

The strength of a synapse can increase or decrease over time—a property that is called synaptic plasticity. Changes in the strength of synapses are thought to underlie learning and memory, and long-lasting changes in synaptic strength involve increases or decreases in the number and size of dendritic spines. Such changes are possible because spines have an internal skeleton that can be assembled and disassembled in a matter of minutes. This 'remodeling' process is regulated by a family of enzymes called small GTPases. One of these, known as Cdc42, has been shown to promote the formation and maintenance of spines in cell culture, but its role in synaptic plasticity, learning and memory remains unknown.

Now, Kim, Wang et al. have used genetically modified mice who have had Cdc42 deleted from excitatory neurons in their forebrain to examine the functions of this enzyme in living animals. These 'knockout' mice showed a small but statistically significant reduction in the number of dendritic spines in the hippocampus. They also showed smaller changes in spine volume and impaired long-term synaptic plasticity in the hippocampus.

When the mice performed long-term memory tests where they learnt to associate a specific set of visual cues with an impending electric shock, the knockout mice performed well for up to a few days. However, when tested again on the same task 45 days later, the knockout mice did not perform as well as normal mice. This is surprising, given the presumed role of long-term synaptic plasticity in learning and memory, and indicates that Cdc42 is required for 'remote memory', the form of memory lasting for many days. Similar results were obtained with another memory test using a water maze, where the animals have to remember the location of a hidden platform. Normal mice remember the location for more than 30 days. In contrast, the knockout mice could only remember the location for a few days.

As well as providing the first demonstration of the role of Cdc42 in synaptic plasticity in live animals, the work of Kim, Wang et al. has provided new insights into the functions of this enzyme in memory. Further work is required to determine how Cdc42 interacts with other proteins present at synapses.

by activating the WASP-family and PAK-family pathways (*Kreis and Barnier, 2009*) during the development of neuronal networks. Little, however, is known about the postnatal roles of Cdc42 in dendritic spines under physiological conditions. Additionally, the effects of deletion of Cdc42 on behavioral characteristics such as learning and memory have not been examined.

In the present study, by using Cdc42 conditional KO mice, we evaluate how postnatal disruption of Cdc42 in excitatory neurons affects structural plasticity of dendritic spines and synaptic plasticity. To confirm the consequential effects of the Cdc42 deletion, we further investigate the spine morphology of knockout neurons and conduct a variety of behavioral tests with the mutant mice.

## Results

### Active Cdc42 is decreased in *Cdc42^{f/f}: Camk2a-Cre* mice

To investigate the neuronal morphology and the behavioral outcomes provoked by postnatal Cdc42 disruption in cortical excitatory neurons in vivo, we crossed *Cdc42^{f/f}* mice with the *Camk2a-Cre* line, which drives Cre recombinase within pyramidal neurons of the forebrain including the hippocampus and the cerebral cortex by p16-p19 (*Tsien et al., 1996*). The total amount of Cdc42 protein in the hippocampus was markedly decreased in *Cdc42^{f/f}: Camk2a-Cre* mice (p120) compared to that of the control littermates as shown in input lanes (*Figure 1A*). The CRIB pulldown assay, which was

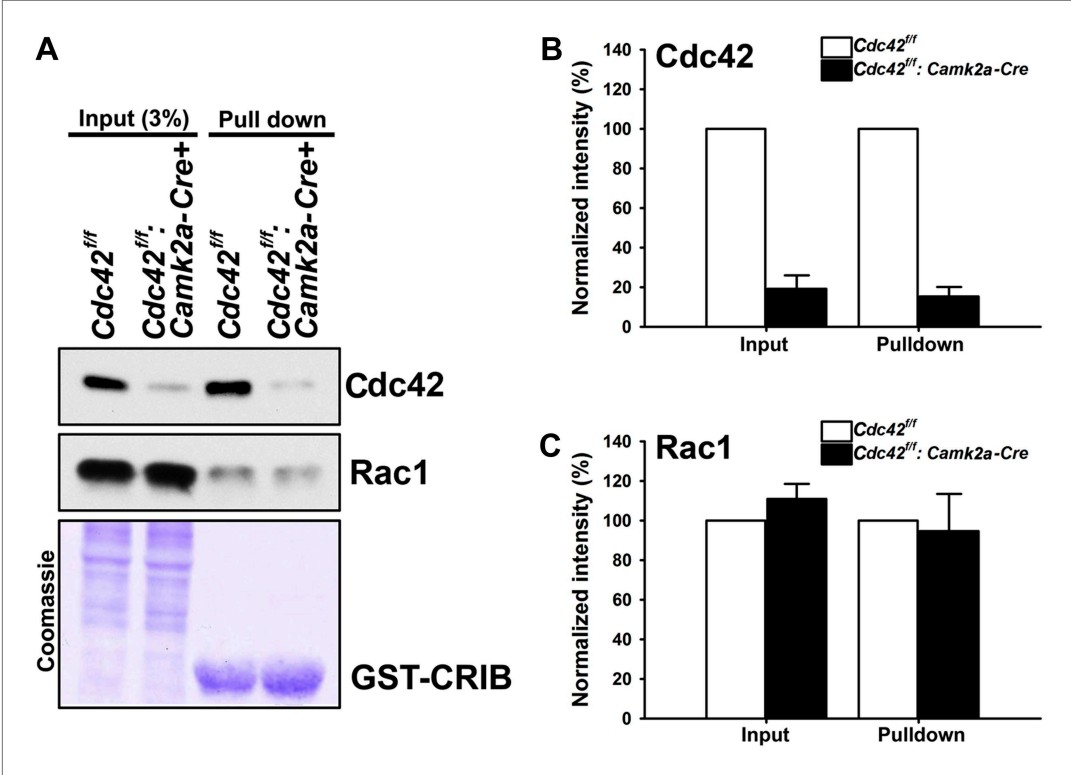

**Figure 1**. Loss of hippocampal Cdc42 in *Cdc42^{f/f}: Camk2a-Cre* mice. (**A**) Top and middle panels, representative western blots of Cdc42 and Rac1 levels for input (left two lanes) and GST-CRIB pulldowns (right two lanes). Bottom panel is a representative coomassie stain showing equivalent amounts of total protein (left two lanes) or GST-CRIB fusion protein (right two lanes). (**B** and **C**) Graphs depicting the quantification of (**B**) Cdc42 or (**C**) Rac1 GTPases from western blot analysis of GST-CRIB pulldowns from hippocampal lysates. n = 4 for each group. *p<0.0001.

performed using with same amounts of hippocampal lysates from p120 mice, clearly displayed a substantial reduction of GTP-bound, active Cdc42 in the mutant mice, whereas, the total and active form levels of Rac1 were similar both in mutants and controls (*Figure 1A*). Coomassie staining confirmed an equal loading of total protein for both groups (*Figure 1A*). Analyses of band intensities from both genotypes revealed that ~80% of total Cdc42 protein was lost in mutant hippocampi compared to those of littermate controls (*Figure 1B*) ($t_{(1,6)}$ = 13.895, *p<0.0001). Because the *Cdc42^{f/f}: Camk2a-Cre* mice used in this study express Cre recombinase selectively in excitatory neurons, the remaining ~20% of Cdc42 protein detected in mutant hippocampi most likely originates from other cell types such as glia or inhibitory interneurons that also express Cdc42 (*Etienne-Manneville and Hall, 2001*).

We next tested whether the loss of total Cdc42 led to a decrease of the active form of Cdc42 and/or a compensatory change in active Rac levels. CRIB pulldown assays revealed an 85% decrease (15% remained) of active Cdc42 proteins in mutant animals (*Figure 1B*) ($t_{(1,6)}$ = 20.725, *p<0.0001). In contrast, there were no changes in total Rac1 and active Rac1 proteins in hippocampi of the mutant mice (*Figure 1C*), suggesting a selective alteration of Cdc42 in excitatory neurons of the mutant hippocampus.

## Selective decrease in dendritic spine density in the hippocampus following postnatal Cdc42 depletion in the forebrain

Cdc42 is known as one of the critical factors regulating dendritic spine morphogenesis in cultured neurons at an early development stage (*Nakayama et al., 2000*). These in vitro observations led us to conduct morphological analyses for KO neurons in vivo to determine whether Cdc42 function in the post-developmental neuron is critical for spine maintenance under physiological conditions, using the *Cdc42^{f/f} : Camk2a-Cre* mice. To analyze the expression pattern of Cre recombinase in the *Camk2-Cre* mouse at the adult stage (p60), the *Camk2a-Cre* mouse was crossed with the *Rosa26-lox-stop-lox-tdTomato* reporter mouse (*Figure 2A*). The cre-induced tdTomato expression was detected in cortical areas

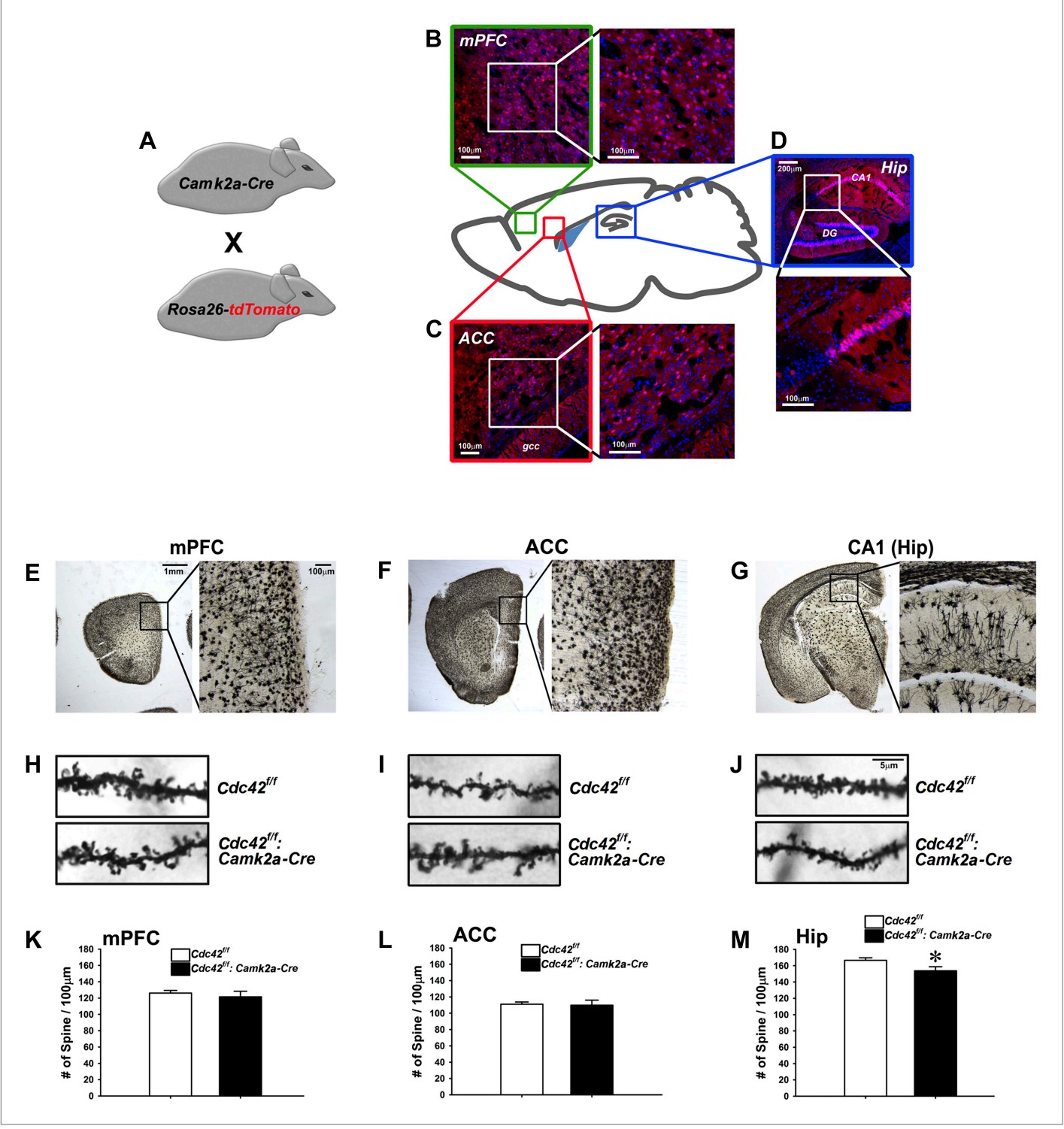

**Figure 2**. Analysis of dendritic spines in *Cdc42^f/f^: Camk2a-Cre* mice. (**A**) Schematic of breeding for the analysis of *Camk2a-cre* expression analysis in **B–D**. (**B–D**) Representative images of cre-dependent tdTomato expression in (**B**) medial pre-frontal cortex (mPFC), (**C**) anterior cingulate cortex (ACC), and (**D**) Hippocampus (Hip). (**E–G**) Representative images of golgi stained tissue sections from (**E**) the mPFC, (**F**) the ACC, and (**G**) CA1 hippocampal region. (**H–J**) Representative images of individual dendritic segments from the (**H**) the mPFC, (**I**) ACC, or (**J**) CA1 hippocampal region (top panels) control *Cdc42^f/f^* or (bottom panels) cKO *Cdc42^f/f^: Camk2a-Cre* mice. (**K–M**) Graphs depicting the quantitative analysis of spines per 100 micron of dendritic segments for each genotype from each region in **H–J**. n = 5 for each group. *p<0.05.

including the medial prefrontal cortex (mPFC) (*Figure 2B*), anterior cingulate cortex (ACC) (*Figure 2C*), and CA1 region of hippocampus (*Figure 2D*). To ascertain the long-term effects of Cdc42 deletion in vivo, we prepared brains from p120 *Cdc42^{f/f}: Camk2a-Cre* mice and conducted Goli-Cox staining (*Figure 2E–G*). Morphological analysis of neurons in hippocampal CA1 revealed a slight (8%), yet statistically significant decrease of spine density in the CA1 pyramidal neurons of *Cdc42^{f/f}: Camk2a-Cre* mice ($t_{(1,8)}$ = 2.447, *$p < 0.05$) (*Figure 2M*). No differences in spine density were found in either the mPFC or the ACC (*Figure 2H,I,K,L*), suggesting that hippocampal neurons are more susceptible to the Cdc42 disruption for maintenance of hippocampal spines.

## Structural and functional synaptic plasticity is abolished by *Cdc42* deletion

Spine enlargement is thought to be the structural basis of LTP and learning and memory (*Matsuzaki et al., 2004*; *Murakoshi, et al., 2011*; *Kim et al., 2013*). Previously we showed that Cdc42 is required for structural and functional plasticity of dendritic spines using shRNA targeted to Cdc42 in rat hippocampus (*Murakoshi et al., 2011*). To further test the roles of Cdc42 in spine plasticity, we transfected neurons of *Cdc42^{f/f}* mice with EGFP-Cre or EGFP (control) together with mCherry as a volume marker using ballistic gene transfer, and measured spine volume change induced by glutamate uncaging in CA1 pyramidal neurons (*Figure 3A–D*). In control neurons, stimulated spines showed sustained volume change lasted more than 30 min (60 ± 19% at 20–30 min) (*Figure 3A,B*). However, in neurons expressing EGFP-Cre, the volume change was significantly impaired (8 ± 5%, $p < 0.05$). In another set of experiments, we expressed mEGFP-Cdc42 together with mCherry and EGFP-Cre. We found that structural plasticity in these neurons was similar to those paired neurons in which EGFP and mEGFP were expressed instead of EGFP-Cre and mEGFP-Cdc42 (64 ± 20% vs 63 ± 23%) (*Figure 3C,D*). Thus, the impaired structural plasticity in neurons expressing EGFP-Cre was caused by removal of Cdc42 and exogenous Cdc42 can rescue the effect. From these experiments, we concluded that Cdc42 is necessary for spine structural plasticity, consistent with our previous report (*Murakoshi et al., 2011*).

To further examine whether Cdc42 is necessary for hippocampal LTP, we measured fEPSP in the CA1 region of hippocampal slices taken from *Cdc42^{f/f}: Camk2a-Cre* mice and their litter-mate *Cdc42^{f/f}* mice at P21-P28 (*Figure 3E*). We found that, while control *Cdc42^{f/f}* mice displayed a robust LTP in response to HFS (49 ± 18% at 30–40 min), LTP induction was significantly impaired in *Cdc42^{f/f}: Camk2a-Cre* mice (−2 ± 9%, $p < 0.05$) (*Figure 3F*). These results indicate that Cdc42 is necessary for hippocampal LTP as well as spine structural plasticity.

## Working memory, locomotor activities, and anxiety levels are normal in *Cdc42^{f/f}: Camk2a-Cre* mice

Previously we showed that proper regulation of actin cytoskeletal remodeling in the forebrain is critical for a variety of behaviors using the *Camk2a-Cre* line to delete ArpC3 (*Kim et al., 2013*). Although prior work clearly shows that Cdc42 is activated downstream of NMDA receptors during LTP, the importance of Cdc42 signaling in postnatal neurons for behavioral responses is unknown. To address this, a battery of behavioral tests was conducted to evaluate the *Cdc42^{f/f}: Camk2a-Cre* mice from the age of p120.

We weighed mice at p120 to check for possible gross health impairments during postnatal development. Neither weight loss nor gain was noted in *Cdc42^{f/f}: Camk2a-Cre* mice when compared to their littermate controls (*Figure 4A*), suggesting normal health and development of cKO mice.

Y-maze spontaneous alteration was analyzed to evaluate the role of Cdc42 forebrain signaling in working memory. The tests revealed that WT and cKO mice engaged in similar numbers of alternations in the 3-way maze, suggesting an intact working memory of Cdc42 cKO (*Figure 4B*).

Locomotor activities were examined by open field test (OFT). In this test, Cdc42 cKO mice displayed normal levels of locomotor (*Figure 4C*) and repetitive activities (*Figure 4D*) that were statistically indistinguishable from their control littermates, demonstrating that cKO mice do no exhibit locomotor behavioral abnormalities. The hippocampus has been reported to play a pivotal role in the processing of emotional information through circuitry connections with other brain regions such as the amygdala (*LeDoux, 2000*; *Engin and Treit, 2007*). Because our *Cdc42^{f/f}: Camk2a-Cre* mice showed defects in the synaptic plasticity and reduction of dendritic spines in the hippocampus, we hypothesized the Cdc42 cKO mice might exhibit an anxiety phenotype. Analysis of the OFT data however revealed that

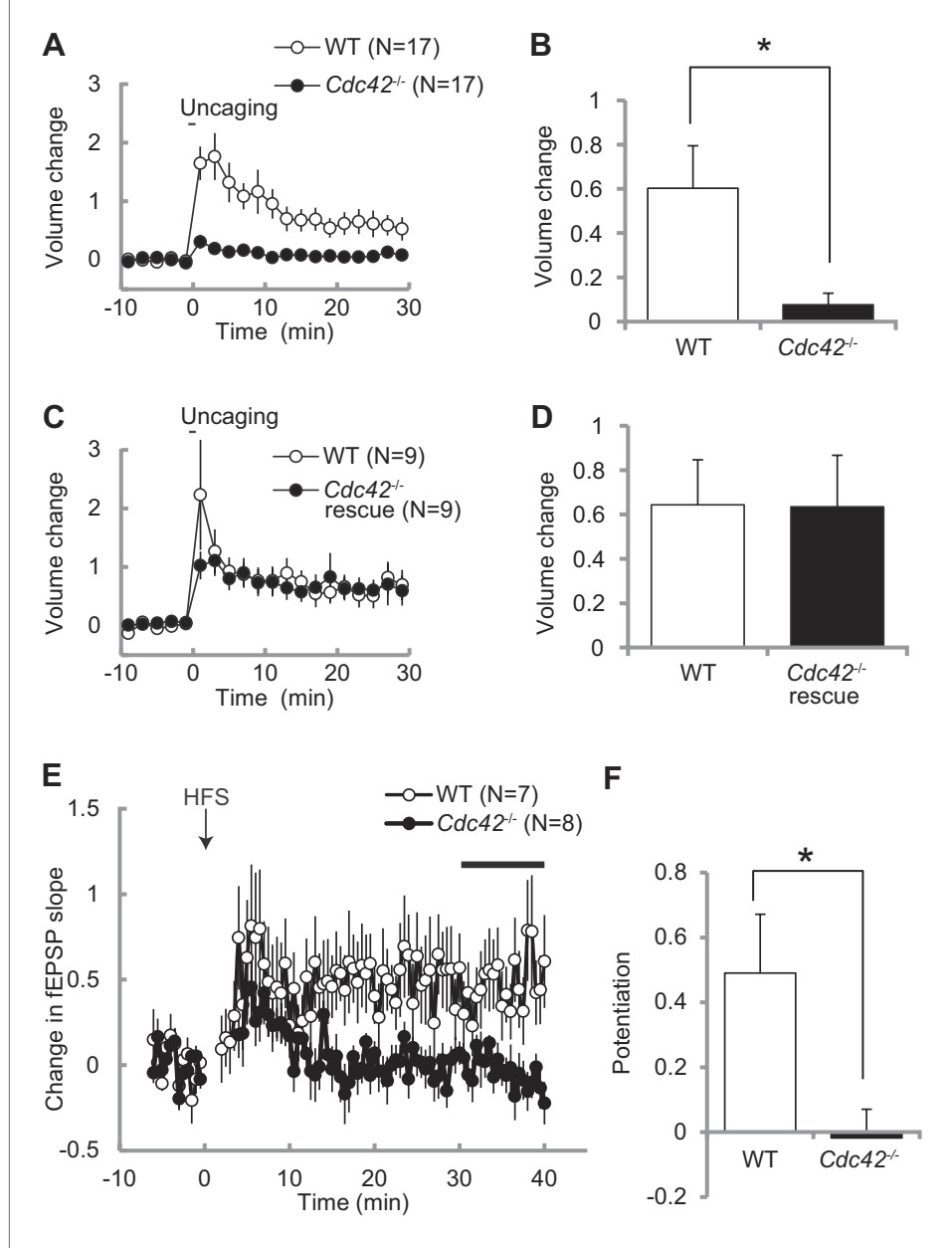

**Figure 3**. Impaired structural and functional synaptic plasticity in *Cdc42^{f/f}: Camk2a-Cre* mice. (**A**) Single spine volume changes in response to glutamate uncaging for (open circle) WT control *Cdc42^{f/f}* or (closed circle) cKO *Cdc42^{−/−}* mice. (**B**) Mean responses for minutes 20–30 between each genotype from (**A**) showing a significant impairment in cKO spines. N = 15 spines/15 slices for each group, *p<0.05. (**C**) Same as in (**A**) except the cKO neurons are co-transfected with a Cdc42 expression construct (rescue). (**D**) Mean responses for minutes 20–30 between WT and cKO *Cdc42^{−/−}* rescue spines are shown. N = 9 spines/9 slices for each group. (**E**) Graph depicting changes in fEPSP slope in response to high-frequency stimulation (HFS; 100 pulses at 100 Hz; three times with 20 s intervals) of the SC–CA1 pathway (time zero). (**F**) Mean fEPSP potentiation for minutes 30–40 was significantly reduced in *Cdc42^{−/−}* hippocampal slices when compared to WT littermates. N = 7 and 8 slices for WT and *Cdc42^{−/−}*, respectively. *p<0.05.

WT and cKO mice spent similar amounts of time in the central and marginal areas of the arena compared with their littermate controls, suggesting normal anxiety levels in the cKO mice (***Figure 4E***).

Anxiety was further evaluated in the elevated zero maze (EZM) test. In this test, cKO mice displayed no significant differences in latencies to the open arms (***Figure 4F***), number of entries to the open

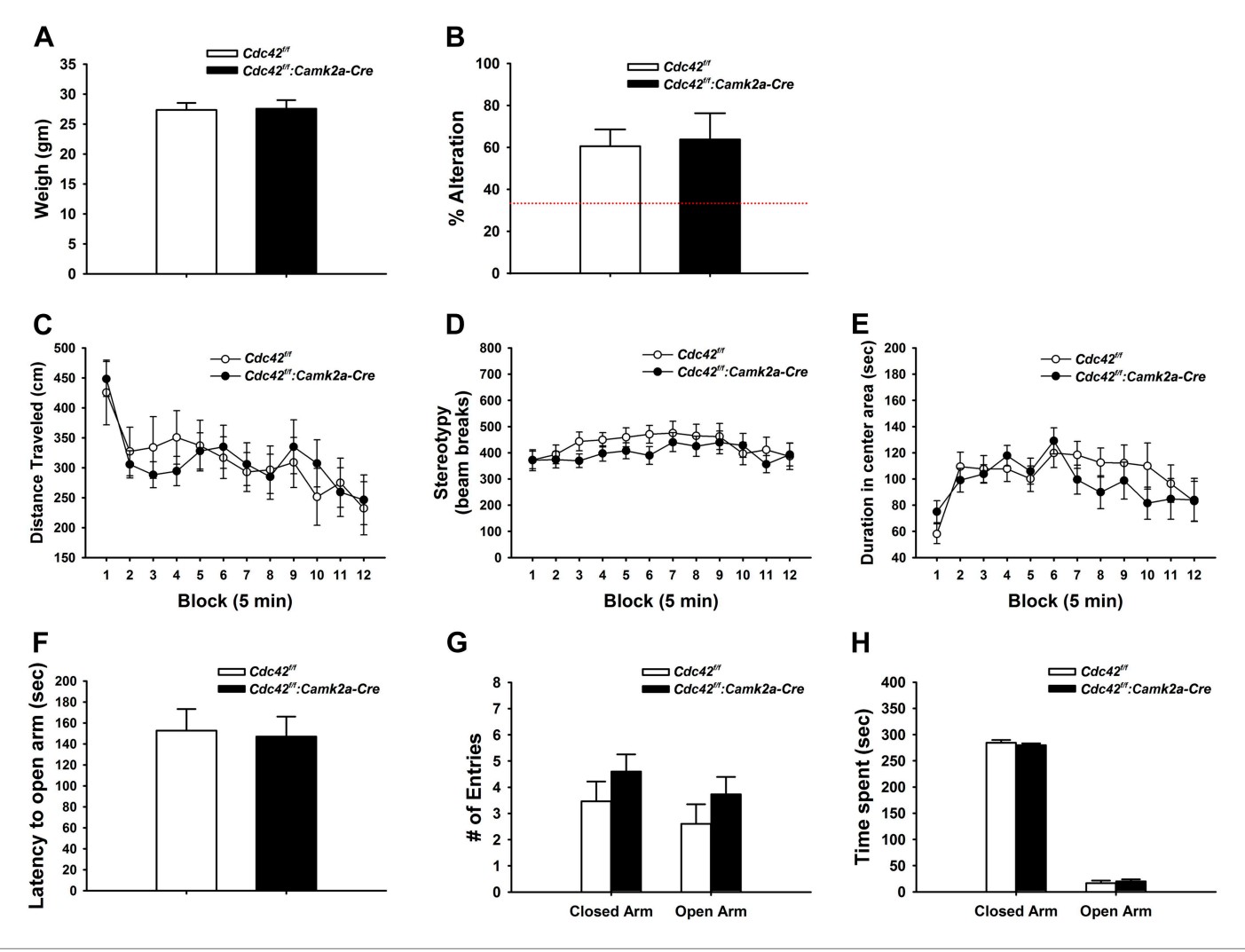

**Figure 4.** Behaviors unaffected by loss of Cdc42. (**A**) Average body weight of *Cdc42*[f/f] (control, open bar) or *Cdc42*[f/f]: *Camk2a-Cre* mice (cKO, black bar). (**B**) Percent alternation in the Y-maze for both genotypes. Dashed line indicates the expected percent correct alternation that would be observed by chance. (**C–E**) Analysis of Open Field exploration behavior for (**C**) distance traveled, (**D**) stereotypy, or (**E**) duration spent in the center of the field. (**F–H**) Analysis of zero maze exploration behavior for (**F**) latency to enter the open arm, (**G**) number of entries to both the closed and open arms of the maze, and (**H**) total time spent in each arm. n = 14 for each group.

arms (*Figure 4G*), and duration in the open arms of the maze (*Figure 4H*) when compared to those of WT controls. Together these data confirmed that the Cdc42 cKO mice show normal anxiety levels.

### *Cdc42*[f/f]: *Camk2a-Cre* mice are deficient in remote memory recall

Synaptic plasticity and neuronal morphology are intimately related to cognitive function (*Martin et al., 2000*; *Segal, 2005*; *Kim et al., 2013*). Based on our findings that Cdc42 deletion leads to defects in structural/synaptic plasticity and spine loss in hippocampus, we also suspected that the behavioral outcomes of *Cdc42*[f/f]: *Camk2a-Cre* mice may be abnormal in certain aspects of hippocampus-dependent cognitive tasks. To test this hypothesis we conducted a variety of behavioral analyses of episodic memory. Contextual memory capability was tested by a fear conditioning paradigm in which mice learn to predict aversive events (mild electric shock) (*Figure 5A*). Following an aversive stimulus in a conditioning chamber, control *Cdc42*[f/f] and cKO *Cdc42*[f/f]: *Camk2a-Cre* mice showed similar freezing rates when placed in the conditioning chamber for the first 4 days (*Figure 5B*). ANOVA with Repeated Measure (RMANOVA) for freezing rates of control *Cdc42*[f/f] and cKO *Cdc42*[f/f]: *Camk2a-Cre*

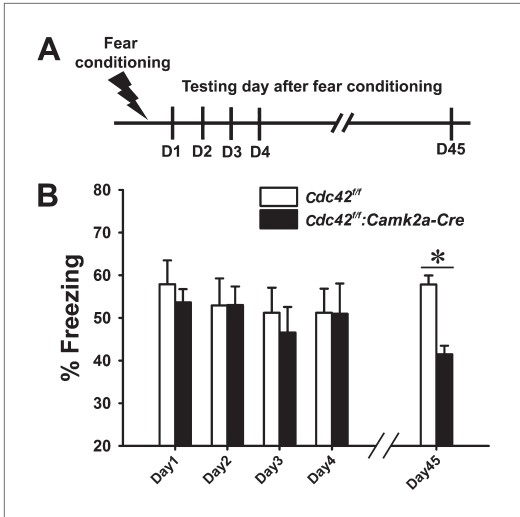

**Figure 5**. Cdc42 cKO mice exhibit reduced memory recall in the fear conditioning learning and memory paradigm. (**A**) Schematic of the fear conditioning protocol in which the mice receive a mild aversive foot-shock on day 1 (D1) in a conditioning chamber. Freezing upon placement in the chamber (without shock) was assessed during acquisition (day 1) or for long-term (days 2–4) or remote memory (day 45). (**B**) Graph depicting the average percent time spent freezing at each time point for each genotype. n = 14 for each group. *p<0.001.

mice revealed no effects of test-day or genotype and no interaction between test-day and genotype. Bonferroni corrected pair-wise comparisons revealed no significant differences between both genotypes from day 1 trough day 4, suggesting that long-term memory retention is not affected in the cKO mice. However, a test of remote memory, which was conducted at 45 days after the conditioning, showed a marked decrease of freezing rates when compared to their WT littermates (*p<0.001; Bonferroni corrected pair-wise comparisons), even though the Cdc42 cKO mice clearly were not impaired in the initial 4 days (*Figure 5B*). These data suggested the Cdc42 cKO mice have pronounced deficits in remote memory recall. Moreover, the normal anxiety level in the cKO mice (*Figure 4E–H*) support the fear conditioning data in that freezing rate (an indicator of memory) was not affected by a difference of innate anxiety levels between each genotype.

We also performed the Morris water maze task to test the spatial memory of the Cdc42 cKO mice (*Figure 6A*). WT and cKO mice explored a similar distance (*Figure 6B*) and expended a similar amount time (data not shown) to find the hidden platform during the 8 days of learning sessions. RMANOVA for the distances explored until reaching the hidden platform for WT and cKO mice revealed significant main effects of test-day ($F_{(7,112)}$ = 28.858, p<0.0001), indicating successful spatial learning (a reduction of swimming distance to reach the hidden platform for each subsequent test day) of both genotypes (*Figure 6B*). However, we found no genotype effect and no interaction between test-day and genotype. Bonferroni corrected pair-wise comparisons found no significant differences between both genotypes from day 1 through day 8. RMANOVA for the time spent to find the hidden platform also revealed a similar result ($F_{(7,112)}$ = 32.793, p<0.0001 [main effect of test-day] for duration, no genotype effect and no interaction between test-day and genotype) (figure not shown). No difference of swim speed was found between WT and cKO mice throughout the acquisition trials (data not shown). These data suggest that cKO mice have normal swimming abilities and spatial memory acquisition capabilities when compared to WT controls. Probe trials conducted every other day during the learning sessions (day 2, 4, 6, 8) showed similar long-term memory performances for both genotypes. ANOVA followed by Bonferroni pair-wise comparisons (among four quadrants) for each probe test revealed that both WT and cKO mice traveled a significantly greater distance in the target quadrant (NE) compared to other three quadrants from day 4 (*Figure 6C,D*) (*ps* < 0.05), indicating a normal capability of long-term memory formation in cKO mice.

Next we performed reversal water maze trainings to test how quickly the mice learned the new location by relocating the platform to the opposite quadrant of the water tank (SW). In this paradigm, cKO mice showed acquisition performances throughout the re-learning sessions (day 9–day 16) that were similar to those of WT controls. RMANOVA for the distances explored and for the time spent to reach the hidden platform revealed significant main effects of test-day for swim distance ($F_{(7,112)}$ = 23.612, p<0.0001) (*Figure 6E*) and for swim duration ($F_{(7,112)}$ = 23.602, p<0.0001) (figure not shown), indicating both genotypes successfully re-learned the new location of the hidden platform. There were no effects of genotype and no interactions between test-day and genotype either in distance and duration, and no significant differences were found between both genotypes as tested by Bonferroni corrected pair-wise comparisons, further indicating normal re-learning capabilities of the cKO mice.

In the reversal probe tests of long-term memory formation, however, Cdc42 cKO mice showed a slight delay in long-term memory formation for the new location compared to their littermate

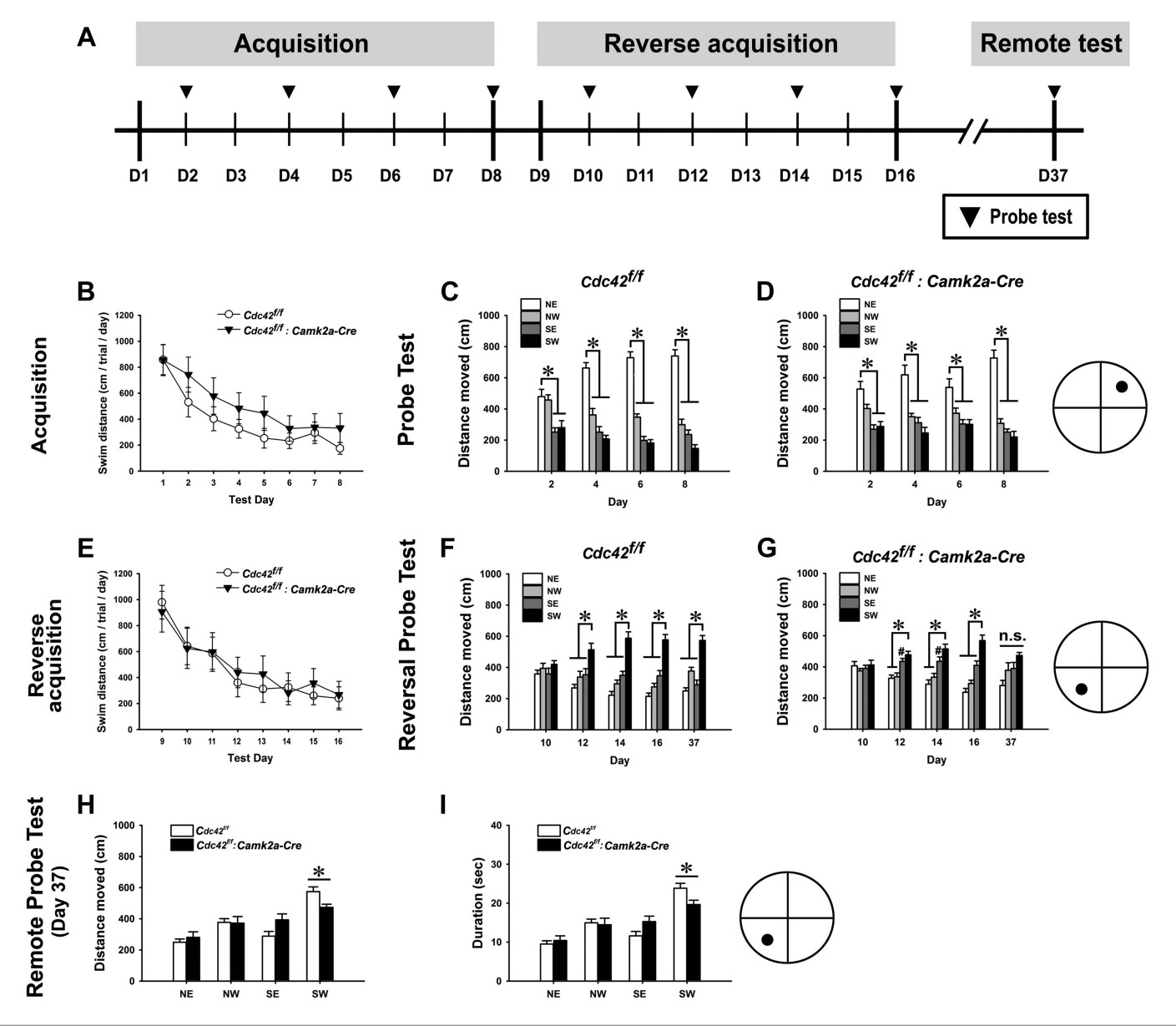

**Figure 6**. Cdc42 is essential for normal memory recall in the water maze test. (**A**) Schematic of the water maze testing schedule showing the acquisition phase during days (**D**) 1–8, platform reversal phase during D9-16, and remote probe trial test on D37. No significant differences were observed between control *Cdc42f/f* and *Cdc42f/f: Camk2a-Cre* cKO mice during water maze acquisition phase as measured by (**B**) total swim distance to the platform, or in distance moved in the target vs non-target quadrants for (**C**) control or *Cdc42* cKO mice. (**E**) Swim distances to the platform during acquisition was also unaffected in the *Cdc42* cKO mice during reversal learning. (**F**) Control mice spent significantly more time in the target vs non-target quadrants during the remote memory probe trial on day 37, however (**G**) *Cdc42* cKO mice did not distinguish between these quadrants. There were significant differences between the control and *Cdc42* mice in both the (**H**) distance moved and (**I**) duration of time spent within the target (SW) quadrant during the remote memory probe trial. n = 8 for WT; n = 10 for cKO. *ps < 0.05, # = no significant difference from target quadrant.

controls (*Figure 6F,G*). ANOVA followed by pair-wise comparisons using Bonferroni post-hoc analyses for each probe test showed that the WT mice traveled a significantly greater distance in the new target quadrant (SW) compared to each of the other three quadrants beginning on day 12 (*Figure 6F*) (*ps < 0.05*). In contrast, the cKO mice did not travel a significantly greater distance in the target vs the adjacent SE non-target quadrant on day 12 or day 14 (*Figure 6G*) (p=0.614 for day 12; p=0.236 for day 14). The cKO mice only distinguished the target from each non-target

quadrant by day 16, as measured by the distance traveled in the target vs each non-target quadrant ($ps < 0.05$).

Probe trials at 21 days after the last reversal probe test (day 37) were next carried out to evaluate the remote memory capability of the cKO mice (bar graphs in right side of each *Figure 6F,G*). In these trials, WT mice traveled more time in target quadrant (SW) compared to other three quadrants. ANOVA revealed significant main effects for distance ($F_{(3,28)} = 29.437$, p<0.0001) (*Figure 6F*) and duration ($F_{(3,28)} = 34.363$, p<0.0001) (figure not shown), suggesting that WT mice successfully recalled the location of the platform. Bonferroni post-hoc tests for each comparisons (SW vs NE, NW, and SE) found that traveling distances and durations of WT mice in the target quadrant (SW) were significantly higher than those of other three quadrants (*ps<0.001 for distance, *ps<0.0001 for duration [figure not shown]). By contrast, cKO mice traveled similar distances and durations among the four quadrants. ANOVA followed by Bonferroni post-hoc tests revealed no main effects and no individual differences among the four quadrants, suggesting a remote memory deficit of cKO mice (*Figure 6G*). We also examined the differences in remote memory between both genotypes. Cdc42 cKO mice traveled significantly shorter distances ($t_{(1,16)} = 3.238$, *p<0.01) (*Figure 6H*) and spent less time ($t_{(1,16)} = 3.208$, *p<0.01) (*Figure 6I*) within the target quadrant compared to those of WT controls (independent *t* tests), again supporting a selective deficit in remote memory. Together, both the fear conditioning and water maze tests indicate that postnatal deletion of Cdc42 in forebrain excitatory neurons does not cause defects in initial memory acquisition or long-term memory retention. Instead the results of these tests show that during reversal learning loss of Cdc42 leads to a slight delay in long-term memory formation and a pervasive deficit in remote memory recall.

## Discussion

In the present study, we analyzed Cdc42 conditional knockout mice using physiological, morphological, and behavioral approaches, and demonstrated that the postnatal disruption of Cdc42 in excitatory neurons of the forebrain leads to a disruption of structural plasticity of dendritic spines, impaired synaptic plasticity, a reduction of the density of dendritic spines in the hippocampus, and a pronounced deficit of remote memory performances in the cKO mice.

### The effects of postnatal Cdc42 disruption on synaptic plasticity

Actin dynamics are known to subserve the activity-dependent morphological alterations of spines, which is necessary for synaptic plasticity (*Fukazawa et al., 2003*). Cdc42 is one of the small GTPase proteins present in the dendritic spine, and is well known for its actin remodeling functions. Because our previous study demonstrated an intensive activation of Cdc42 specifically within the spine head following single spine glutamate uncaging stimuli (*Murakoshi et al., 2011*), we expected that Cdc42 cKO neurons may show specific defects in activity-dependent spine enlargement. In line with this hypothesis, glutamate uncaging-induced spine plasticity was markedly disrupted in both the transient and sustained phase of spine enlargement upon postnatal deletion of Cdc42. Electrophysiological LTP with *Cdc42^f/f: Camk2a-Cre* mice also confirmed a plasticity deficit upon Cdc42 deletion, indicating that Cdc42 activation is pivotal for both activity-dependent morphogenesis of spines and in the functional synaptic plasticity. These results suggest that Cdc42 exerts its morphing activities not only in the developing neurons (neuronal migration and establishment of polarity etc) but also in mature neurons.

### Spine morphology in *Cdc42^f/f: Camk2a-Cre* mice

Since we found a robust defect of glutamate uncaging-inducing spine enlargement in the Cdc42 cKO mice, and because Cdc42 is generally believed to play a key role in the morphogenesis of variety cells by regulating actin structure, we expected a substantial effect of Cdc42 deletion on the maintenance of spine density following long-term exposure to cre-recombinase within forebrain excitatory neurons. However, in contrary to this hypothesis, the spine loss in cKO neurons was very mild: only 8% reduction in spine density was found in hippocampal pyramidal neurons. Moreover, there was no detectable change in spine density in medial prefrontal cortex or anterior cingulate cortex of the cKO mice. This stands in contrast to our prior findings in *ArpC3^f/f: Camk2a-Cre* mice, which exhibit a loss of approximately 50% of spines in both the hippocampus and medial prefrontal cortex (*Kim et al., 2013*). Comparing these results suggest that although *Camk2a-Cre* drives the loss of floxed alleles in both regions, Cdc42 signaling is specifically involved in rapid time-scale spine morphing/maintaining processes during the neuronal activation but, unlike Arp2/3, it is not strongly involved in the long-term maintenance of spine

morphology in mature neurons. This observation suggests functional characteristics of Cdc42 strongly dependent on the developmental status of neuron. During the early developmental stages, Cdc42 is reported to be essential for the morphogenesis and polarity establishment of neurons that leads to a permanent change of cell shapes and locations (*Mueller, 1999*; *Luo, 2000*; *Wong et al., 2001*; *Aoki et al., 2004*; *Schwamborn and Puschel, 2004*; *Garvalov et al., 2007*). However, during the postnatal period, Cdc42 activity may be functionally restricted to manage activity-dependent changes of actin dynamics that governs aspects of synaptic plasticity rather than gross maintenance of spines.

## Behavioral alterations by postnatal Cdc42 deletion

Even though the hippocampi of *Cdc42ᶠ/ᶠ: Camk2a-Cre* mice showed robust deficits in both structural and functional measurements of synaptic plasticity, the cKO mice surprisingly displayed normal performances in many behavioral tests, including those for working memory and memory acquisition. Remote memory as determined by both the fear conditioning and water maze tests, however, was significantly diminished in the Cdc42 cKO mice. How does the loss of Cdc42 lead to the deficit in the process of remote memory? Two explanations are possible to account for the remote memory defect of the cKO mice: a memory storage defect, or a remote memory retention and retrieval problem. Normal performances in learning and intact working memory of the cKO mice suggest that memory formation and storage processing appears not to be altered by Cdc42 deletion, although it should be noted we did observe a slight delay in long-term memory formation that was specific to the water maze reversal test. We speculate the retention or the retrieval capacities for the remote memory are most likely affected in *Cdc42ᶠ/ᶠ: Camk2a-Cre* mice.

Notably, a line of studies have found that hippocampal lesions impair recent memory such as short- and long-term memories, whereas the lesions do not affect remote memory (*Squire and Alvarez, 1995*; *Knowlton and Fanselow, 1998*). This 'graded retrograde amnesia' supports an idea that remote memory retrieval may be independent of hippocampal functions. This finding is opposite to the behavioral phenotype observed in *Cdc42ᶠ/ᶠ: Camk2a-Cre* mice. However, other studies using human patients who have medial temporal lobe (MTL) damages find these patients exhibit a memory loss without any temporal gradient (*Rosenbaum et al., 2001*; *Moscovitch et al., 2006*; *Winocur et al., 2010*), revealing that the retrograde amnesia is not always graded. Moreover, a recent study showed the hippocampus is tightly involved in the retrieval of remote memory. Remote memory recall was affected by a temporal and precise optogenetic inhibition of CA1 excitatory neurons in hippocampus (*Goshen et al., 2011*). If Cdc42 is critically involved in the neuronal activation during the process of remote memory retrieval, Cdc42 loss may result in a remote memory defect with normal learning and working memory.

Rac and Cdc42 share similar downstream pathways: both Rac and Cdc42 can remodel actin via the p21 Kinase (PAK) and LIMK pathway to inactivate cofilin-mediated actin severing or by stimulating the WAVE1/ N-WASP pathway to activate Arp2/3 dependent actin polymerization. Both pathways are implicated in synaptic plasticity and are thought to be critical for processes important for learning and memory. Thus it has remained unclear as to whether Rac or Cdc42 can be distinguished from each other at the level of behavioral phenotypes in intact animals. It is of interest, therefore, to compare the results reported here with the analogous *Rac1ᶠ/ᶠ: CamKIIα-Cre* mice previously characterized (*Haditsch et al., 2009*). Loss of Rac1 results in impaired hippocampal LTP similar to our findings in the Cdc42 cKO mice. Surprisingly, however, the behavioral impairments are quite different. Loss of Rac1 results in impaired working memory, but has no effect on long-term or remote memory. This is in stark contrast to the Cdc42 cKO mice which display normal working memory, but are impaired in remote memory. Together, these findings reveal that although the biochemical pathways that modulate actin remodeling evoked by synaptic activation of Rac1 and Cdc42 may overlap, their functions during learning and memory are clearly distinguishable from the other, with little overlap. Further work will be required to define how the physiologic relevance of Rac1 and Cdc42 are functionally segregated during these processes despite their similar biochemical pathways.

## Materials and methods

### Animals

Conditional Cdc42 knockout animals were generously provided by Dr Cord Brakebusch (University of Copenhagan). Cdc42 conditional knockout mice (*Cdc42ᶠ/ᶠ*) were crossed with the *Camk2a-Cre* line

(stock no. 005359; The Jackson Laboratory; Bar Harbor, ME) for synaptic plasticity studies, biochemical assays, Golgi-cox staining, and behavioral tests. To analyze the expression pattern of Cre recombinase, the *Rosa26-lox-stop-lox-tdTomato* reporter line (generously provided by Dr Fan Wang) was crossed with the *Camk2a-Cre* line. Littermate male and female mice from heterozygous pairings were used in all experiments. All mice were housed in the Duke University's Division of Laboratory Animal Resources facilities and all procedures were conducted as approved by the Duke University Institutional Animal Care and Use Committee in accordance with National Institutes of Health guidelines.

### Spine structural plasticity

Glutamate uncaging was performed as described (*Murakoshi et al., 2011*). P6 *Cdc42$^{f/f}$* pups were deeply anesthetized with isoflurane and decapitated; the hippocampus was rapidly dissected into medium containing (mM): HEPES 25, NaHCO$_3$ 2, Sucrose 248, glucose 10, KCl 4, MgCl$_2$ 5, CaCl$_2$ 1. Then, 350 µm slices were cut with a tissue chopper (Ted Pella, Inc.; Redding, CA) and transferred to the surface of membrane inserts (EMD Millipore; Darmstadt, Germany), and placed in culture media containing (mM): L-glutamine 1, CaCl$_2$ 1, MgSO$_4$ 2, D-glucose 12.9, NaHCO$_3$ 5.2, Na-HEPES 30, insulin 0.001, Ascorbic acid 0.53, 20% heat-inactivated horse serum, 80% HEPES-based MEM 8.4 g/l (pH 7.35, 320 Osm). Slice-containing plates were maintained in a 37°C incubator with 5% CO$_2$. 5–10 days after incubation, cultures were transfected biolistically with a gene gun. To make bullets, 40–50 µg DNA were mixed with 8–12 mg 1.6 µm gold particles (Bio-Rad; Hercules, CA). Amount of DNA used was: 20 µg EGFP + 20 µg mCherry (Control); 20 µg EGFP-Cre + 20 µg mCherry (Cdc42 knockout); 20 µg EGFP + 20 µg mCherry + 10 µg monomeric EGFP (mEGF) (Control for rescue); 20 µg EGFP-Cre + 20 µg mCherry + 10 µg mEGFP-Cdc42 (rescue). Uncaging experiments were performed 3–4 days after transfection.

A Ti:Sapphire laser was tuned to 720 nm to uncage the caged glutamate in artificial cerebral spinal fluid (ACSF) that contained (mM): NaCl 130.0, KCl 2.5, NaHCO$_3$ 2.0, NaH$_2$PO$_4$ 1.25, glucose 25.0, CaCl$_2$ 4.0, tetrodotoxin 0.001 and MNI-caged L-glutamate 2.0 at 25–27°C. Structural plasticity associated with LTP was induced by 4–7 ms, 5–8 mW uncaging pulses applied at 0.5 Hz for 30 pulses. Spines of neurons expressing mCherry and EGFP were imaged with a 1030 nm ytterbium-doped laser (Amplitude). EGFP-Cre expression was confirmed with the presence of strong fluorescence in the nucleus. Volume change was monitored by measuring the change in the intensity of EGFP or mCherry fluorescence over time.

### LTP

Hippocampal slices, 400 µm thick, were prepared from *Cdc42$^{f/f}$: Camk2a-Cre* mice and their litter-mate control *Cdc42$^{f/f}$* mice (P21-P28). Field EPSP (fEPSP) was measured with a pipette filled with 1 M NaCl located at the dendritic level (~100 µm away from the somatic layer), while stimulating axons with a bipolar tungsten electrode located at striatum radiatum (single pulses, 50–100 µs, 30 s intervals). Stimulation strength was adjusted so that fEPSP amplitude is less than 50% saturation. LTP was induced with three sets of high frequency stimulation (HFS; 100 Hz, 1 s, 20 s intervals). The experiments were performed in artificial CSF (ACSF) containing 2 mM MgCl$_2$ and 2 mM CaCl$_2$ at room temperature. Persons who performed experiments and analyses were blinded from genotypes until the whole experiments and analyses are finished and statistical significance was calculated.

### Golgi-Cox staining

Golgi-Cox staining procedures were performed as described (*Kim et al., 2013*). Mice were deeply anesthetized with isoflurane and then transcardially perfused with 4% PFA. Brains were removed and treated with solutions A and B from the FD Rapid GolgiStain Kit (FD Neuro Technologies, Columbia, MD) for 2 weeks, and then treated with solution C for 7 days. Sections (100 µm thick) were cut by cryostat and transferred to solution C and incubated for 24 hr at 4°C. After brief rinsing with distilled water, floating sections were stained consecutively with solutions D and E for 30 min and then transferred to a 0.05% gelatin solution. Sections were mounted on glass slides, dehydrated through a graded series of ethanol concentrations, and then mounted with Permount. Images were collected by a MicroPublisher 5.0 MP color camera (QImaging; Surrey, BC, Canada) on a Zeiss Axio Imager microscope under a 100 × oil-immersion objective using MetaMorph 7.6.5 software. For quantification, spine density from segments of secondary or tertiary branches of CA1 pyramidal neurons in the stratum radiatum of the hippocampus were measured.

## Cdc42 activity assay and western blotting

Cdc42 activity was measured using glutathione S-transferase (GST)-Cdc42 and Rac1 interactive binding domain (CRIB) binding assay as described (*Kim et al., 2009*). *Cdc42^{f/f}: Camk2a-Cre* and control *Cdc42^{f/f}* mice were deeply anesthetized with isoflurane and hippocampi were rapidly removed, homogenized with 15 strokes using a Teflon-glass homogenizer in ice-cold lysis buffer (mM): NaCl 150, HEPES 25 (pH 7.4), EDTA 1, EGTA 0.5, 0.5% NP-40 that contained, protease/phosphatase inhibitors and then sonicated. The homogenate was centrifuged at 10000×g for 5 min at 4°C and then the supernatant was collected. The protein concentration was determined with the Bradford protein assay (Bio-Rad).

For GST-CRIB binding assay, GST protein fused to CRIB domain from human PAK1B was expressed in *E.coli* BL-21 cells and used for the assay. Identical amounts of purified GST-CRIB proteins were pre-incubated with 20 μl of glutathione-Sepharose beads (4 Fast Flow; Amersham Biosciences; Pittsburgh, PA) in NETN buffer (20 mM Tris–HCl, pH 8.0; 100 mM NaCl, 1 mM EDTA, 0.5% NP-40), then washed twice each with 600 μl of NETN and 600 μl of lysis buffer. The hippocampal lysates were incubated with the GST-CRIB protein-bound glutathione-Sepharose beads for 2 hr at 4°C on a rotator (15 rpm). The beads were collected by centrifugation (1500×g/1 min), and the supernatant was removed and the pellet was rinsed with lysis buffer. The bound proteins were eluted from the beads by boiling the samples in SDS loading buffer (1 M Tris–HCl [pH 6.8], 10% [vol/vol] SDS, 50% [vol/vol] glycerol, 5% [vol/vol] 2-mercaptoethanol, and 1% [vol/vol] bromophenol blue).

For Western blotting, 10 μg of samples were electrophoresed through 12% SDS-PAGE (Bio-Rad) and transferred onto a nitrocellulose membrane (Whatman; Pittsburgh, PA), and nonspecific sites were blocked with 5% nonfat dry milk in TRIS-buffered saline (TBS; pH 7.4) containing 0.05% Tween-20. For detection, the membranes were probed with rabbit anti-Cdc42 polyclonal antibody (Santa Cruz; Dallas, TX) and mouse anti-Rac1 monoclonal antibody (BD; San Jose, CA) for 24 hr at 4°C. After washing, the membranes were incubated with horseradish peroxidase-conjugated secondary antibodies (GE Healthcare Life Sciences; Pittsburgh, PA) for 2 hr, washed, and then developed using the ECL system (Thermo Scientific; Waltham, MA). Membranes were then exposed to autoradiography films (Genesee Scientific; San Diego, CA).

## Contextual fear conditioning

Fear conditioning was conducted as described (*Porton et al., 2010*). Med-Associates mouse fear conditioning chambers were used for conditioning and testing. The tests consist of three sessions: conditioning (day 1), long-term fear memory tests (day 2-day 5), and remote fear memory test (day 45). Following a 2 min acclimation in the conditioning chamber, mice received a 0.4 mA scrambled foot shock for 2 s. Each mouse remained in the chamber for an additional 30 s before being placed into its home cage. Fear memory testing was conducted daily for 4 days by placing the mice in the conditioning chamber for 5 min in the absence of foot shock. The remote memory was tested at 45 days after the conditioning. Freezing rate was analyzed by trained observers who were blind to the genotypes of mice using Noldus Observer (Noldus Information Technology; Wageningen, Netherlands) software. Freezing was defined by criteria previously described for mice as the absence of all visible movement except that required for respiration (*Anagnostaras et al., 2000*).

## Morris water maze

Morris water maze task was conducted as described (*Porton et al., 2010*). A 120 cm diameter water tank was used. Opaque water in the tank was maintained at 25°C. The water pool was divided into four quadrants (NE, NW, SE, and SW). A 12 cm diameter round platform was submerged 1 cm below the water surface and 20 cm apart from the wall of the water tank at the NE quadrant. Testing consisted of three sessions: acquisition and probe trials (day 1-day 8), reversal acquisition and reversal probe trials (day 9-day 16), and remote probe trials (day 37). 1 week prior to testing, all mice were handled daily for 5 min and then were placed in a pan of shallow water (1 cm) for 30 s to acclimate them to water. On the seventh day after handling, each mouse was placed onto the hidden platform in the NE quadrant for 20 s and then allowed to swim freely for 60 s before being returned to the platform for 15 s. Acquisition testing consisted of 32 trials given across 8 days with four trials administered per day. Trials were run in pairs, with each pair separated by 60 min. Probe trials were conducted without platform at the end of days 2, 4, 6 and 8. Reversal acquisition and reversal probe tests were conducted same way as the acquisition/probe tests described above, but the platform location was moved from NE to SW. Remote test was conducted at day 37. For each trial, the release point for the animals was

randomized across seven equally spaced points along the perimeter of the maze. All test trials were 1 min in duration. The swim distance, swim time were determined by Noldus Ethovision (Noldus Information Technology).

### Y-maze

Y-maze test was performed as described (*Kim et al., 2013*). Spontaneous alternation in a Y-maze was conducted under indirect illumination (80–90 lux) in a 3-arm Y-maze. The mouse was placed into the center arm of the maze and permitted free exploration for 5 min. Entry into an arm was defined as the mouse being more than 1 body length into that arm, with both hind-paws past the entrance to that arm. An arm alternation was defined as three successive entries into each of the different arms. Alternation, calculated as the total number of alternations divided by the total number of arm entries minus 2, was expressed as a percentage.

### Open field test (OFT)

OFT was performed as described (*Kim et al., 2013*). Mice were placed into an open field (AccuScan Instruments) and their activities were monitored over 1 hr under 350 lux illumination using VersaMax software (AccuScan Instruments; Columbus, OH). Locomotor (distance traveled), rearing (vertical beam-breaks), stereotypical activities (repetitive beam-breaks <1 s), and anxiety level (duration in center area of arena) were measured in 5-min time-bins.

### Elevated zero maze (EZM)

EZM was performed as described (*Pogorelov et al., 2005*). The zero maze is a 5.5 cm-wide circular running platform elevated 43 cm from the floor. The inside diameter of the maze is 34 cm with two opposite quadrants were enclosed by 11 cm-high walls. Mice were placed into a closed quadrant and permitted to investigate the maze for 5 min under 50–60 lux illumination. The behaviors were recorded and analyzed by Noldus Observer (Noldus Information Technology). The scored behaviors included percent of time spent in open quadrants and total number of transitions between quadrants, and latency to enter the open quadrants.

### Statistical analyses

All data are expressed as means ± SEM and all statistics were analyzed using SPSS software (SPSS 20). Independent $t$ tests were used for analysis of differences between two groups. When comparing more than two groups, ANOVA followed by Bonferroni post-hoc analyses was used. To monitor changes over time, repeated-measures ANOVA (RMANOVA) were run followed by Bonferroni corrected pair-wise comparisons. A $p < 0.05$ was considered statistically significant.

## Acknowledgements

We thank Cord Brakebusch for providing Cdc42 cKO mice, Fan Wang for *Rosa26-lox-stop-lox-tdTomato* reporter mice, Clifford Heindel, Junsu Kang, and Airon Wang for technical support and David Kloetzer for lab management. This study was supported by NIH (MH103374, NS059957 R01MH080047, R01NS068410, R01MH095090) and HHMI.

## Additional information

### Funding

| Funder | Grant reference number | Author |
| --- | --- | --- |
| National Institutes of Health | R01NS068410 | Ryohei Yasuda |
| Howard Hughes Medical Institute | | Ryohei Yasuda |
| National Institutes of Health | R01MH080047 | Ryohei Yasuda |
| National Institutes of Health | R01NS059957 | Scott H Soderling |
| National Institutes of Health | R01MH103374 | Scott H Soderling |
| National Institutes of Health | R01MH095090 | Ryohei Yasuda |

The funders had no role in study design, data collection and interpretation, or the decision to submit the work for publication.

## Author contributions

IHK, HW, SHS, Conception and design, Acquisition of data, Analysis and interpretation of data, Drafting or revising the article; RY, Conception and design, Analysis and interpretation of data, Drafting or revising the article

## Ethics

Animal experimentation: The institutional animal care and use committee (IACUC) of the Duke University Medical Center and the approved animal protocols are A113-11-05 and A288-11-11. The institutional guidelines for the care and use of laboratory animals were followed.

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
