## [Decision Letter]

Thank you for sending your work entitled “Loss of Cdc42 Leads to Defects in Synaptic Plasticity and Remote Memory Recall” for consideration at *eLife*. Your article has been favorably evaluated by a Senior editor and 3 reviewers, one of whom is a member of our Board of Reviewing Editors.

The Reviewing editor and the other reviewers discussed their comments before we reached this decision, and the Reviewing editor has assembled the following comments to help you prepare a revised submission.

Kim and colleagues examined the role of Cdc42 in synaptic plasticity in mature neurons and in behaviors such as learning and memory in adult animals. Consistent with previous results using RNAi, the authors show that postnatal loss of Cdc42 interferes with LTP-induced spine structural enlargements in juvenile animals. Using field recordings, they further show that these structural deficits are accompanied by functional deficits – a complete loss of LTP in Schaffer collateral pathway. Surprisingly, despite these dramatic changes in plasticity in juvenile animals, they found only a small, but significant, decrease in spine density in the hippocampus in adults and no change in spine density in the PFC. Remarkably, adult animals show a dramatic and selective impairment of remote memory following postnatal loss of Cdc42, despite that locomotor activity, memory acquisition, anxiety level and working/long-term memories are preserved.

This is a well-written written and carefully executed study that addresses an interesting and important question on the role of Cdc42 once neurons have matured. Overall, the data are convincing, and the experiments are well-planned. A specific deficit in remote memory is fascinating and future studies with these animals could significantly advance our understanding of the mechanisms of remote memory retention/retrieval.

Major comments:

1) The authors show selective decrease in hippocampal spine density following Cdc42 depletion in the forebrain. They further show that structural plasticity in spines is impaired after cdc42 deletion. This is interesting however major questions remain open: are these changes in structural plasticity in spines underlie the impairments in functional synaptic plasticity (LTP) and behavior? If so is the permanent decrease in dendritic spine density or the lack in spine plasticity responsible for such impairments?

An experiment that will considerably contribute to answering these questions will be to examine whether stimulation leading to LTP or behavioral training lead to changes in spine structure and density. If so whether there is a different between the cdc42 depleted mice and wt mice and whether over expression of the cdc42 in cdc42 KO mice can rescue LTP and behavior and the impairments in changes in spine morphology? Alternatively, although less ideally, the abstract and conclusions should be revised to better match the data (i.e. suggesting at most a 'possible' or 'plausible' link).

2) It is not clear why the authors concentrate only on hippocampal LTP and spine morphology and density. They show in their behavioral results that remote memory, but not short- and long-term memories, is impaired in cdc42 depleted mice. However, whereas recent memories depend on the hippocampus remote memory storage and retrieval are mediated by the cortex. Specific cortical sites are activated by remote memory processes and such activation is absent in mice with remote memory deficits (e.g. Frankland et al., 2004). Furthermore, inactivation of the anterior cingulate cortex disrupt contextual fear memory at remote, but not recent time points showing that this cortical site is required for remote memory.

Since short- and long-term memories are intact in the cdc42 depleted mice but in contrast remote memory is impaired it would be important to examine the effects of the absence of cdc42 on synaptic and structural plasticity in cortex in addition to the hippocampus since the behavioral impairments indicate also on cortical functional impairments. This would considerably contribute to the understanding of the cellular and anatomical brain contribution to the memory deficits that are observed in the study.

3) In Figure 2, the authors show Cdc42 deletion leads to a reduction in the number of dendritic spines in the hippocampus but not in the medial prefrontal cortex. Measuring mEPSCs in these regions would further support the conclusion because functional measures are often more sensitive than structural analysis. If any other cortical areas are found to be important in additional experiments performed for the comment #2, for instance, the anterior cingulate cortex, these regions should also be measured of spine density and mEPSCs.

4) The description of the results in Figure 6 is confusing. It is not clear what is significant for main effects of test-day for swim distance in Figure 6 (p<0.0001, as stated in the text). In addition, Day 12 and 14 look different in the Cdc42 KO than in WT (Figure 6) – is long-term memory really normal in the KO at these time points? This section should be clarified.

---

## [Author Response]

*1) The authors show selective decrease in hippocampal spine density following Cdc42 depletion in the forebrain. They further show that structural plasticity in spines is impaired after cdc42 deletion*. *This is interesting however major questions remain open: are these changes in structural plasticity in spines underlie the impairments in functional synaptic plasticity (LTP) and behavior? If so is the permanent decrease in dendritic spine density or the lack in spine plasticity responsible for such impairments?*

We agree that the link between LTP, structural plasticity, and an animal’s learning and memory is not directly addressed in this paper. These are the questions that have been asked in the field for the past decades and we believe that this paper significantly improves our knowledge about these questions.

*An experiment that will considerably contribute to answering these questions will be to examine whether stimulation leading to LTP or behavioral training lead to changes in spine structure and density. If so whether there is a different between the cdc42 depleted mice and wt mice and whether over expression of the cdc42 in cdc42 KO mice can rescue LTP and behavior and the impairments in changes in spine morphology? Alternatively, although less ideally, the abstract and conclusions should be revised to better match the data (i.e. suggesting at most a 'possible' or 'plausible' link)*.

It has been known that LTP-inducing stimulus can cause spine volume increase and new spine formation. However, imaging spines in the hippocampus of live animals before and after learning are still challenging without removing a significant portion of the cortex. Also, it is believed that only a small fraction of spines contributed to learning of one specific context in the hippocampus. Thus, with currently available techniques, it is extremely difficult to answer these questions. Also, rescuing synaptic and behavioral phenotypes in Cdc42 KO by re-expressing exogenous Cdc42 via virus is a great idea but technically challenging for large structures such as the hippocampus and negative results will not add any insights.

Therefore, following an alternative suggestion that was kindly provided by the referees, we addressed these issues by removing the claim of the causality link between impaired synaptic plasticity and loss of remote memory in the abstract and conclusions as follows:

“… We found that deletion of Cdc42 impaired LTP in the Schaffer collateral synapses and postsynaptic structural plasticity of dendritic spines in CA1 pyramidal neurons in the hippocampus. Additionally, loss of Cdc42 did not affect memory acquisition, but instead significantly impaired remote memory recall. Together these results indicate that the postnatal functions of Cdc42 may be crucial for the synaptic plasticity in hippocampal neurons, which contribute to the capacity for remote memory recall.” (Abstract)

“In the present study, we analyzed Cdc42 conditional knockout mice using physiological, morphological, and behavioral approaches, and demonstrated that the postnatal disruption of Cdc42 in excitatory neurons of the forebrain leads to a disruption of structural plasticity of dendritic spines, impaired synaptic plasticity, a reduction of the density of dendritic spines in the hippocampus, and a pronounced deficit of remote memory performances in the cKO mice.” (Discussion)

*2) It is not clear why the authors concentrate only on hippocampal LTP and spine morphology and density. They show in their behavioral results that remote memory, but not short- and long-term memories, is impaired in cdc42 depleted mice. However, whereas recent memories depend on the hippocampus remote memory storage and retrieval are mediated by the cortex. Specific cortical sites are activated by remote memory processes and such activation is absent in mice with remote memory deficits (e.g. Frankland et al., 2004). Furthermore, inactivation of the anterior cingulate cortex disrupt contextual fear memory at remote, but not recent time points showing that this cortical site is required for remote memory*.

Since short- and long-term memories are intact in the cdc42 depleted mice but in contrast remote memory is impaired it would be important to examine the effects of the absence of cdc42 on synaptic and structural plasticity in cortex in addition to the hippocampus since the behavioral impairments indicate also on cortical functional impairments. This would considerably contribute to the understanding of the cellular and anatomical brain contribution to the memory deficits that are observed in the study.

We thank the reviewers for the insightful suggestions. We performed additional experiments to test the expression pattern of Cre recombinase in our *CamKIIα*-Cre line in the hippocampus and neocortex by crossing *CamKIIα*-Cre mice with the *Rosa26-loxP-tdTomato-loxP* reporter mice (Figure 2). We indeed found that some neurons in the anterior cingulate cortex (ACC) show significant expression (Figure 2). We further performed Golgi staining in ACC to measure spine density. However, we found no difference in spine density between wildtype and Cdc42 KO in the ACC, suggesting that the role of Cdc42 in spine morphology of the ACC is minimal. In the revised manuscript, we described these findings in the text as:

“To analyze the expression pattern of Cre recombinase at the adult stage (p60), the CamKllα-Cre mouse was crossed with the *Rosa26-lox-stop-lox-tdTomato* reporter mouse (Figure 2). The cre-induced tdTomato expression was detected in cortical areas including the medial prefrontal cortex (mPFC) (Figure 2), anterior cingulate cortex (ACC) (Figure 2), and CA1 region of hippocampus (Figure 2).”

“No differences in spine density were found in either the mPFC or the ACC (Figure 2).”

*3) In*
Figure 2*, the authors show Cdc42 deletion leads to a reduction in the number of dendritic spines in the hippocampus but not in the medial prefrontal cortex. Measuring mEPSCs in these regions would further support the conclusion because functional measures are often more sensitive than structural analysis. If any other cortical areas are found to be important in additional experiments performed for the comment #2, for instance, the anterior cingulate cortex, these regions should also be measured of spine density and mEPSCs*.

We agree that measurement of mEPSCs would confirm a decrease of functional spines in the hippocampus. However, at least in our hands, detecting 8% reduction of change in the number of synapses with mEPSCs frequency is not possible (for example, please see error bars of Szatmari et al., 2013, *J. Neurosci*). Also, as we discussed above, we found that anterior cingulate cortex (ACC) also express Cre, but did not find any difference in spine density between control mice and Cdc42 KO mice.

*4) The description of the results in*
Figure 6
*is confusing. It is not clear what is significant for main effects of test-day for swim distance in*
Figure 6
*(p<0.0001, as stated in the text)*.

We thank the reviewers for this comment. The significant main effect of Figure 6 means the swim distances until reaching the hidden platform for both WT and cKO was significantly reduced throughout the acquisition trials for 8 days, indicating that both genotypes are able to learn the new location during the acquisition phase of the reversal test.

We clarified the description of the statistical analyses in the revised manuscript as:

*“*Next we performed reversal water maze trainings to test how quickly the mice learned the new location by relocating the platform to the opposite quadrant of the water tank (SW). In this paradigm, cKO mice showed acquisition performances throughout the re-learning sessions (day 9-day 16) that were similar to those of WT controls. RMANOVA for the distances explored and for the time spent to reach the hidden platform revealed significant main effects of test-day for swim distance [F(7,112) = 23.612, p<0.0001] (Figure 6) and for swim duration [F(7,112) = 23.602, p<0.0001] (figure not shown), indicating both genotypes successfully re-learned the new location of the hidden platform. There were no effects of genotype and no interactions between test-day and genotype either in distance and duration, and no significant differences were found between both genotypes as tested by Bonferroni corrected pair-wise comparisons, further indicating normal re-learning capabilities of the cKO mice.*”*

*In addition, Day 12 and 14 look different in the Cdc42 KO than in WT (*Figure 6*) – is long-term memory really normal in the KO at these time points? This section should be clarified*.

We thank the reviewers for their insights about Figure 6. Indeed, Cdc42 KO mice show significantly slower long-term memory formation compared to WT on Day 12 and 14. We clarified this point in the revised manuscript as follows:

“In the reversal probe tests of long-term memory formation, however, Cdc42 cKO mice showed a slight delay in long-term memory formation for the new location compared to their littermate controls (Figure 6). ANOVA followed by pair-wise comparisons using Bonferroni post-hoc analyses for each probe test showed that the WT mice traveled a significantly greater distance in the new target quadrant (SW) compared to each of the other three quadrants beginning on day 12 (Figure 6) (ps<0.05). In contrast, the cKO mice did not travel a significantly greater distance in the target versus the adjacent SE non-target quadrant on day 12 or day 14 (Figure 6) (p=0.614 for day 12; p=0.236 for day 14). The cKO mice only distinguished the target from each non-target quadrant by day 16, as measured by the distance traveled in the target versus each non-target quadrant (ps<0.05).”